# Multimodal Instruction Tuning with Hybrid State Space Models

## Abstract

Handling lengthy context is crucial for enhancing the recognition and understanding capabilities of multimodal large language models (MLLMs) in applications such as processing high-resolution images or high frame rate videos. The rise in image resolution and frame rate substantially increases computational demands due to the increased number of input tokens. This challenge is further exacerbated by the quadratic complexity with respect to sequence length of the self-attention mechanism. Most prior works either pre-train models with long contexts, overlooking the efficiency problem, or attempt to reduce the context length via downsampling (e.g., identify the key image patches or frames) to decrease the context length, which may result in information loss. To circumvent this issue while keeping the remarkable effectiveness of MLLMs, we propose a novel approach using a hybrid transformer-MAMBA model to efficiently handle long contexts in multimodal applications. Our multimodal model can effectively process long context input exceeding 100k tokens, outperforming existing models across various benchmarks. Remarkably, our model enhances inference efficiency for high-resolution images and high-frame-rate videos by about 4 times compared to current models, with efficiency gains increasing as image resolution or video frames rise. Furthermore, our model is the first to be trained on low-resolution images or low-frame-rate videos while being capable of inference on high-resolution images and high-frame-rate videos, offering flexibility for inference in diverse scenarios.

## 1 Introduction

Recent efforts have extended large language models (LLMs) to incorporate multiple modalities, leading to breakthroughs in various multimodal tasks (Liu et al., 2024d;c;b; Luo et al., 2024a; Chen et al., 2023b). Despite these advances, existing MLLMs still struggle with tasks that require long input sequence, e.g. granular visual recognition (Tong et al., 2024), high frame rate videos and long videos. For example, many well-trained models (e.g., GPT-4V) produce hallucinations when identifying small and occluded objects in images (Tong et al., 2024) while the same also happens to video MLLMs (Ma et al., 2023), a limitation that hinders the practical application of MLLMs.

Various methods have been explored to improve multimodality processing in different domains by incorporating lengthy context, including increasing the resolution of input images (Liu et al., 2024c; Tong et al., 2024; Hong et al., 2024; Li et al., 2023a; Wang et al., 2023; Luo et al., 2024b), increasing frame rate of input videos and increasing frame sampling rate for long videos. Although better performance is achieved by incorporating extra lengthy context, increasing context length directly can significantly escalate computational demands. For instance, increasing the resolution of images to 448×448 pixels raises the computational complexity of models like LLaVA by approximately 1.4 times compared to the default 336×336 resolution. Moreover, pre-trained vision encoders in current MLLMs typically do not support long contexts due to their fixed context length (Liu et al., 2024d;b; Alayrac et al., 2022), making training unstable with significantly increased resolution. Consequently, most prior works have pre-trained models with long contexts (Li et al., 2023a), overlooking efficiency issues. Besides, some works divide the high-resolution images into smaller patches and then encode them independently (Liu et al., 2024b;c), which might lose the spatial information and also lead to quadratic computational cost increase. Additionally, some methods shorten the context length (Hong et al., 2024; Luo et al., 2024b) by inject high-resolution image

features into the hidden layers of LLMs, which leads to varying degrees of visual information loss and thus reduces effectiveness.

To address these issues, our work proposes using a hybrid transformer-mamba model (e.g., Jamba (Lieber et al., 2024)) to overcome the quadratic complexity associated with transformer models. The hybrid architecture enables our multimodal model to efficiently process the long context resulting from high resolutions and frame rates without compromising effectiveness. For instance, it operates $4\times$ faster than current open-source models (e.g. LLaVA-Next-13B) when the resolution is 4368*4368. In addition, we also propose a train-on-short-infer-on-long recipe, enabling our model to be trained on inputs with a short context (e.g. low-resolution images) for better training efficiency and do inference on inputs with longer contexts (e.g. high-resolution images) for better performance. This approach addresses both the efficiency and effectiveness problems simultaneously. To summarize, our contributions include:

- We introduce a hybrid transformer-mamba multimodal model, MMJAMBA, optimized for efficiently processing lengthy contexts from high resolutions and frame rates. Our "train-on-short-infer-on-long" strategy allows the model to train on short-context inputs (e.g., low-resolution images) for efficiency and infer on long-context inputs (e.g., high-resolution images) for enhanced performance.

- We conducte experiments on both images and videos across 18 benchmark datasets focusing on images and videos. Our results show that MMJAMBA achieves state-of-the-art performance compared to open-source and proprietary models, consistently outperforming LLaVA-NeXT (Liu et al., 2024c) and Gemini Pro 1.0 (Team et al., 2023), occasionally matching or surpassing proprietary models like GPT-4V (OpenAI, 2023). More importantly, we show that our model achieves the best efficiency when processing high-resolution images and high frame rate videos.

- We conduct comprehensive model analyses, including ablation and case studies, to elucidate the inner workings and demonstrate the model's performance in real-world scenarios.

## 2 RELATED WORKS

**Multimodal Large Language Models.** Most current multimodal large language models (MLLMs) integrate large language models (LLMs) with vision encoders, such as ViT (Dosovitskiy, 2020) and CLIP (Radford et al., 2021), by incorporating image representations into the LLMs (Liu et al., 2024d;b;c). The LLM then performs various vision-language (VL) tasks in an autoregressive manner. Within this framework, MLLMs primarily differ in their approaches to combining text and image representations. Most works employ a modular architecture that utilizes an intermediate network to map visual features into the text token embedding space of the LLM. Prominent examples, such as LLaVA (Liu et al., 2024d), often use a multi-layer perceptron (MLP) layer to link the visual encoder with the LLM. Alternatively, other approaches use sampler-based modules to bridge the visual encoder and the LLM (Li et al., 2023c; Dai et al., 2023). While these sampler-based modules effectively reduce the number of visual tokens, they typically require large-scale pre-training to achieve satisfactory performance.

**MLLMs with High-Resolution Images.** MLLMs often use pre-trained visual encoders for vision-dependent tasks. However, these encoders typically rely on low resolutions, such as $224 \times 224$ or $336 \times 336$ pixels, which limits their ability to perceive small or blurry objects. This limitation can lead to failures in tasks that require clear and recognizable details for fine distinctions between objects, such as OCR and document understanding (Tong et al., 2024). Recently, various methods have been introduced to incorporate high-resolution inputs to enhance the capabilities of MLLMs by providing more fine-grained visual features (Luo et al., 2023a; Hong et al., 2024; Li et al., 2023a; Liu et al., 2024c; Tong et al., 2024; Wang et al., 2023). Notable examples include LLaVA-Next (Liu et al., 2024c), LLaVA-HR (Luo et al., 2024b), OtterHD (Li et al., 2023a), InternLM-XComposer2-4KHD (Dong et al., 2024), and InternVL 1.5 (Chen et al., 2024b). Among them, many methods (Hong et al., 2024; Liu et al., 2024b;c) divide high-resolution images into smaller patches and encode them independently. These patches are subsequently concatenated for further processing. While this approach can enhance performance, it also results in a quadratic increase in computational cost due to the extended context length.

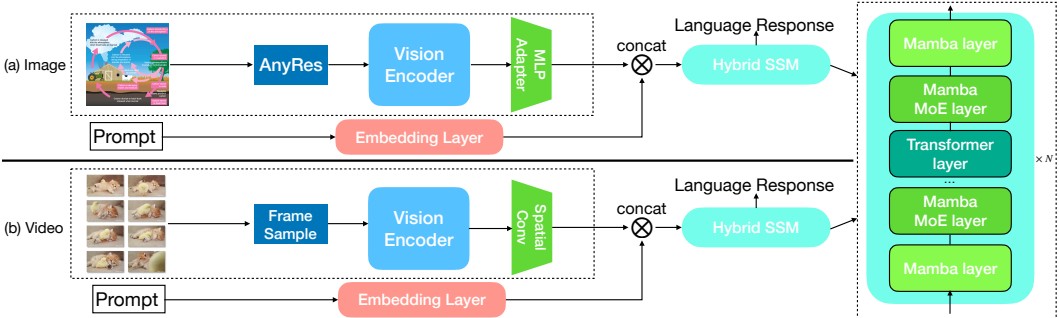

Figure 1: Overview of MMJAMBA. Image representations of all divided patches are flattened and concatenated. The hybrid state space model consists of interleaved Mamba and Transformer layers.

**MLLMs for Videos.** Several works, such as VideoChat (Li et al., 2023d) and Video-LLaMA (Zhang et al., 2023), have introduced LLM-based multimodal models for end-to-end, chat-centric video comprehension. However, these models can only process a limited number of frames, restricting their ability to handle long videos. To tackle the computational challenges posed by the large number of visual tokens in long videos, models like mPLUG-Video (Li et al., 2022) and LLaMA-VID (Li et al., 2023e) have been proposed, mainly relying on visual token reduction, which can result in loss of visual information.

## 3 METHOD

### 3.1 PRELIMINARIES

Traditional structured State Space Models (SSMs) (Gu et al., 2021; Gu et al.) describe the evolution of system states through time, defined as $(\Delta, A, B, C)$. $A \in \mathbb{R}^{N \times N}$ describes how current states evolve into next states, $B \in \mathbb{R}^{N \times 1}$ determines how external inputs influence the evolution, $C \in \mathbb{R}^{1 \times N}$ transforms hidden system states to observed measurements, and $\Delta$ is a time-scale parameter that helps transform $A$ and $B$ into discrete-time parameters $\bar{A}$ and $\bar{B}$. The discretized form of the state space model can be obtained as:

$$\bar{A} = \exp(\Delta A) \tag{1a}$$

$$\bar{B} = (\Delta A)^{-1}(\exp(\Delta A) - I) \cdot \Delta B \tag{1b}$$

$$h_t = \bar{A}h_{t-1} + \bar{B}x_t \tag{1c}$$

$$y_t = Ch_t \tag{1d}$$

$$K = \left(C\bar{B}, C\bar{A}\bar{B}, \dots, C\bar{A}^k\bar{B}, \dots\right) \tag{1e}$$

$$y = x * K \tag{1f}$$

Based on the above structured SSM, the selective SSM–Mamba (Gu & Dao, 2023) is further introduced to endow the model with the ability to selectively propagate or forget information according to the sequential input tokens. Specifically, the selective SSM achieves the content-aware reasoning by introducing input-dependent coefficients matrix $B(x)$ and $C(x)$, as well as the parameter $\Delta(x)$.

Despite advances at handling long-distance relationships, Mamba still falls short in comparison with similarly sized Transformer models. Jamba (Lieber et al., 2024) combines the strengths of both by stacking Transformer layers and Mamba layers in a specific ratio, improving efficiency and performance over the original Transformer and Mamba models.

### 3.2 ARCHITECTURE

MMJAMBA adopts the standard MLLM architecture, comprising a visual encoder, an MLP adapter, and a backbone LLM. Detailed design and implementation of the components are provided below. Figure 1 illustrates the architecture.

**Vision Encoder** Our vision encoder is responsible for encoding images or videos. We adopt the AnyRes approach (Liu et al., 2024c) to process images with varying aspect ratios. Specifically, we allow a maximum of 4 tiles during training. Consequently, this set includes all 8 possible combinations of aspect ratios formed by 1 to 4 tiles, such as {1:1, 1:2, 2:1, 2:2, 3:1, 1:3, 1:4, 4:1}. Then, each image is independently encoded to a sequence of visual features by a transformer-based Visual Encoder. For videos, we adopt a consistent frame sampling approach that extracts a fixed number of frames from each video. The extracted frames are then fed into the visual encoder. We use CLIP-ViT-Large (Radford et al., 2021), with the final layer removed, as the vision encoder. During inference, we adopt an AnyFrame mechanism which samples any number of frames from the input videos for further encoding. More details are in the appendix (See A.1).

**Modality Adapter** The adapter is a simple learnable module that aligns the feature of vision and text by transforming the dimension of the original visual representation to the dimension of the tokens of language model. We train separate adapters for the image understanding task and the video understanding task.

**LLM Backbone** We employ a hybrid transformer-mamba model as the LLM backbone. In particular, we adopt, Jamba, a hybrid decoder architecture that mixes Transformer layers with Mamba layers. The largest pretrained Jamaba model leverages a mixture-of-experts (MoE) architecture, scaling up to 52 billion parameters with 12 billion active. We chose this model for the following reasons: 1) To the best of our knowledge, it is the largest public available model with a hybrid transformer-MoE architecture at the time of doing this work; 2) it demonstrates strong performance on language understanding and reasoning benchmarks; 3) The MoE architecture makes it more computationally efficient compared to models of similar scale.

## 3.3 Model Training

We have two training stages, with the vision encoder remaining frozen throughout.

**1st stage - Adapter Pretraining**: Following (Liu et al., 2024d), we perform modality alignment between images (or videos) and text by pre-training the adapter. All other components except for the adapter are frozen in this stage.

**2nd stage - Visual Instruction Fine-Tuning**: Here, we activate all parameters except the vision encoder. The model is trained by minimizing the causal language modeling loss: $l = -\sum_{i=1}^{|y|} \log p_\theta(y_i|\hat{y}_{1:i-1}, q)$, where $\theta \leftarrow (\theta^{LLM}, \theta^v, \theta')$ represents the model's trainable parameters, $y_i$ is the ground-truth target, and $\hat{y}_{1:i-1}$ denotes the i-1 preceding tokens of the output $y_i$.

**Train-Short-Inference-Long** Due to the recurrent nature of Mamba layer in our Jamba LLM backbone, our model could adopt a long context length during inference while using a short context length during training, which could achieve good efficiency at both training stage and inference stage. Therefore, for image understanding, we use low resolution images during training for shorter context length and thus better training efficiency. During inference, we use high resolution images for longer context length, which contains more fine-grained visual information. For video understanding, we use a low resolution and lower frame number during training for better training efficiency. During inference, we use a high resolution and higher frame number for longer context length, allowing to capture more fine-grained and richer visual information.

## 4 Image Understanding Experiments

### 4.1 Experimental Setup

**Implementation Details** With our train-low-inference-high recipe, we use a maximal resolution of 672*672 during training, corresponding to 2304 tokens. Therefore, the maximal sequence length is set to 4096 during training. During inference, we use different maximal resolutions including 672*672, 1344*1344 and 2688*2688, corresponding to 2880, 9792 and 37440 visual tokens respectively. Therefore, the maximal sequence length is set to 4k, 12k and 40k respectively. We employ the pre-trained CLIP ViTL as the vision encoder, a two-layer MLP as the vision-language adapter, and Jamba-52B as the LLM. All the training processes were conducted for one epoch using the AdamW (Loshchilov, 2017) optimizer and a cosine learning rate schedule, without further tuning. The learn-

Table 1: Comparison with state-of-the-art methods on comprehensive zero-shot benchmarks. Act. means the number of activated parameters.

| Methods | Act. | MME | MMB-EN | MM-Vet | LLaVA-Wild | SEED-IMG | MMMU-val |
|---|---|---|---|---|---|---|---|
| 7B to 13B Models | | | | | | | |
| InstructBLIP (Dai et al., 2023) | 7.9B | 36.0 | 23.7 | - | 60.5 | - | - |
| Qwen-VL-Chat (Bai et al., 2023) | - | 1487.5 | 60.6 | - | - | 58.2 | - |
| LLaVA-v1.5 (Liu et al., 2024b) | 7.1B | 1510.7 | 64.3 | 30.5 | 63.4 | 66.1 | - |
| LLaMA-VID (Li et al., 2023e) | - | 1564.1 | 61.5 | - | - | 59.1 | - |
| VILA (Lin et al., 2023b) | 7.1B | 1533.0 | 68.9 | 34.9 | 69.7 | 61.9 | - |
| SPHINX-Intern2 (Liu et al., 2024a) | - | 1206.4 | 57.9 | 36.5 | 68.8 | - | 35.5 |
| LLaVA-NeXT (Liu et al., 2024c) | 7.6B | 1589.7 | 68.7 | 47.3 | 83.2 | 72.2 | 35.3 |
| Mini-Gemini (Li et al., 2024c) | 7.3B | 1523 | 69.3 | 40.8 | - | 36.1 | 31.4 |
| MM1 (McKinzie et al., 2024) | - | 1529.3 | 79.0 | 42.1 | 81.5 | 69.9 | 37.0 |
| LLAVA-NeXT (Liu et al., 2024c) | 13B | 1575 | 70 | 48.4 | 87.3 | 71.9 | 35.3 |
| MoE Models | | | | | | | |
| SPHINX-MoE (Liu et al., 2024a) | 13.5B | 1485.3 | 71.3 | 40.9 | 70.2 | 73.0 | 31.1 |
| Mini-Gemini (Li et al., 2024c) | 13.5B | 1639.5 | 75.6 | 45.8 | - | - | 41.8 |
| CuMo (Li et al., 2024a) | 13.5B | 1639.5 | 75.3 | 48.7 | 84.7 | 73.2 | 45.0 |
| LongLLaVA (Wang et al., 2024) | 13B | 1630.1 | 72.6 | 40.5 | - | 68.9 | 42.1 |
| MMJAMBA | 12.4B | 1655 | 80.9 | 51.6 | 83.9 | 72.8 | 44.4 |
| Private Models | | | | | | | |
| GPT4V (OpenAI, 2023) | - | 77.0 | 74.4 | - | - | 56.8 | 49.9 |
| Gemini 1.5 Pro (Team, 2024) | - | 74.3 | 74.3 | - | - | 58.5 | 52.1 |
| Claude 3 Opus | - | 63.3 | 59.2 | - | - | 59.4 | 50.5 |
| Qwen-VL-Max (Bai et al., 2023) | - | 1790.1 | 77.6 | 66.6 | - | - | 51.4 |

Table 2: Comparison with state-of-the-art methods on Vision-Language tasks. Act. means the number of activated parameters.

| Methods | Act. | SQA-IMG | TextVQA ($VQA^T$) | GQA | POPE | VQA v2 | Vizwiz |
|---|---|---|---|---|---|---|---|
| 7B to 13B Models | | | | | | | |
| InstructBLIP (Dai et al., 2023) | 7.9B | 60.5 | 50.1 | 49.2 | - | 60.9 | 34.5 |
| Qwen-VL-Chat (Bai et al., 2023) | - | 68.2 | 61.5 | 57.5 | - | 78.2 | 38.9 |
| LLaVA-v1.5 (Liu et al., 2024b) | 7.1B | 66.8 | 58.2 | 62.0 | 85.9 | 78.5 | 50.0 |
| LLaMA-VID (Li et al., 2023e) | - | 65.2 | 51.1 | 56.4 | - | - | - |
| VILA (Lin et al., 2023b) | 7.1B | 68.2 | 64.4 | 62.3 | 85.5 | 79.9 | - |
| SPHINX-Intern2 (Liu et al., 2024a) | - | 59.1 | 38.1 | 56.2 | - | 55.7 | - |
| LLaVA-NeXT (Liu et al., 2024c) | 7.6B | 72.8 | 65.7 | 64.8 | 87.3 | 82.2 | 57.6 |
| Mini-Gemini (Li et al., 2024c) | 7.3B | 65.2 | 51.3 | - | - | - | - |
| MM1 (McKinzie et al., 2024) | - | 72.6 | 72.8 | - | 86.6 | 82.8 | - |
| LLAVA-NeXT (Liu et al., 2024c) | 13B | 73.6 | 67.1 | 65.4 | 86.2 | 82.8 | 60.5 |
| MoE Models | | | | | | | |
| SPHINX-MoE (Liu et al., 2024a) | 13.5B | 74.5 | 68.0 | 63.8 | 89.6 | 81.1 | 61.9 |
| Mini-Gemini (Li et al., 2024c) | 13.5B | - | 69.2 | - | - | - | - |
| CuMo (Li et al., 2024a) | 13.5B | 77.9 | 66.0 | 63.8 | 85.7 | 81.8 | - |
| LongLLaVA (Wang et al., 2024) | 13B | 75.9 | - | 62.2 | - | - | - |
| MMJAMBA | 12.4B | 77.3 | 70.7 | 64.7 | 87.8 | 82.6 | 57.6 |
| Private Models | | | | | | | |
| GPT4V (OpenAI, 2023) | - | - | 78.0 | - | - | - | - |
| Gemini 1.5 Pro (Team, 2024) | - | - | 73.5 | - | - | - | - |
| Claude 3 Opus | - | - | 76.0 | - | - | - | - |
| Qwen-VL-Max (Bai et al., 2023) | - | - | 79.5 | - | - | - | - |

ing rate is set to 1e-3 for pre-training the MLP adapter and reduced to 7e-6 for visual instruction tuning. All experiments were performed on 32 A100 GPUs with an accumulative batch size of 256.

**Training Datasets** For two training stages of our model , we utilize high-quality data to enhance cross-modality understanding and generation. This dataset includes LLaVA-558K (Liu et al., 2024d) for modality alignment during MLP adapter pretraining and LLaVA665K (made up of LLaVA-Instruct-158K (Liu et al., 2024d), ShareGPT-40K (Chen et al., 2023a), VQAv2 (Antol et al., 2015), GQA (Hudson & Manning, 2019), OKVQA (Marino et al., 2019), OCRVQA (Mishra et al., 2019), A-OKVQA (Schwenk et al., 2022), RefCOCO (Yu et al., 2016) and VG Krishna et al. (2017)),

Table 3: Main Results on Multiple-Choice Video QA (MC-VQA) and Open-Ended Video QA (OE-VQA) benchmarks. Act. means the number of activated parameters.

| Methods | Act. | EgoSchema | Perception | MVBench | VideoMME | MSVD | ActivityNet |
|---|---|---|---|---|---|---|---|
| Proprietary Models | | | | | | | |
| Gemini 1.0 Pro (Team et al., 2023) | - | 55.7 | 51.1 | - | - | - | 49.8/- |
| Gemini 1.5 Pro (Team, 2024) | - | 63.2 | - | - | 75.7 | - | 56.7/- |
| GPT4V (OpenAI, 2023) | - | 55.6 | - | 43.7 | 60.7 | - | 59.5/- |
| GPT4O (OpenAI, 2024) | - | 72.2 | - | - | 66.2 | - | 61.9/- |
| Open-source Models | | | | | | | |
| LLaMA-VID (Li et al., 2023e) | 7B | 38.5 | 44.6 | 41.9 | 25.9 | 69.7/3.7 | 47.4/3.3 |
| Video-LLaVA (Lin et al., 2023a) | 7B | 38.4 | 44.3 | 41.0 | 40.4 | 70.7/3.9 | 45.3/3.3 |
| VideoChat2 (Li et al., 2024b) | 7B | 42.2 | 47.3 | 51.1 | 33.7 | 70.0/3.9 | 49.1/3.3 |
| LLaVA-NeXT-Video Liu et al. (2024c) | 7B | 43.9 | 48.8 | 46.5 | 33.7 | 67.8/3.5 | **53.5/3.2** |
| VideoLLaMA2 Cheng et al. (2024) | 7B | 50.5 | 49.6 | 53.4 | 44.0 | 71.7/3.9 | 49.9/3.3 |
| VideoLLaMA2-Mixtral Cheng et al. (2024) | 13.5B | 53.3 | 52.2 | 53.9 | 48.4 | 70.5/3.8 | 50.3/3.4 |
| LongLLaVA (Wang et al., 2024) | 13B | - | - | 54.6 | 51.7 | - | - |
| MMJAMBA | 12.4B | **58.7** | **55.8** | **61.0** | **50.1** | **73.7/4.1** | 48.6/3.5 |

ShareGPT4V (Chen et al., 2023a), LAION-GPT4V[1], DocVQA (Mathew et al., 2021), AI2D (Kembhavi et al., 2016), ChartQA (Masry et al., 2022), DVQA (Kafle et al., 2018) and ALLaVA-Instruct-4V (Chen et al., 2024a) (about 1.68 million single- or multi-round conversations) for visual instruction fine-tuning.

**Evaluation Benchmarks** To show our model's effectiveness, we evaluated our model on zero-shot multimodal benchmarks, including SEED (Li et al., 2023b) (Image), MMB (Liu et al., 2023) (MMBench), MME (Fu et al., 2024a), MM-Vet (Yu et al., 2023), MMMU (Yue et al., 2024), and MathVista (Lu et al., 2023) datasets. Additionally, we reported results on well-known visual question answering datasets, such as $VQA^T$ (TextVQA), GQA (Hudson & Manning, 2019), VQA v2 (Antol et al., 2015), VizWiz (Gurari et al., 2018), and SQAI (Lu et al., 2022) (ScienceQA-Image).

## 4.2 MAIN RESULTS

**Comprehensive Multimodal Benchmarks** In Table 1, we compare our approach with previous leading open-source and closed-source methods across various comprehensive zero-shot multimodal benchmarks. These benchmarks assess the model's visual understanding, reasoning, multidisciplinary abilities, and even logical thinking and math capabilities. Overall, we observe a competitive performance to even better performance compared with the open-source models with the same and even slightly larger parameter size, confirming the effectiveness of our proposed method. It should be noted that our model outperforms the LongLLaVA, a concurrent work, (Wang et al., 2024) by a large margin on all the benchmarks, which further shows the effectiveness of our model.

**Visual Question Answering Benchmarks** In Table 3, we also present a comparison of MM-JAMBA with existing methods on widely used visual question answering benchmarks. Datasets such as TextVQA (VQAT) require the model to have certain OCR (Optical Character Recognition) capabilities to read and reason over the text and scene in the given images. Similarly, noticeable performance improvements can be observed on the five datasets compared to baselines with the same and even larger number of activated parameters. In particular, the performance increases using MMJAMBA on TextVQA and GQA are considerably significant, demonstrating its ability to handle distinct details from images by incorporating granular visual information from high-resolution images.

## 5 VIDEO UNDERSTANDING EXPERIMENTS

### 5.1 EXPERIMENTAL SETUP

**Implementation Details** With our train-short-inference-long recipe, we use a frame number of 8 during training. During inference, we use different frame numbers including 8, 16, 32 and 64. We employ the pre-trained CLIP ViT as the vision encoder, a 2D convolution layer as the vision-language adapter, and Jamba-52B as the LLM. All the training processes were conducted for one

---
[1]https://huggingface.co/datasets/laion/gpt4v-dataset

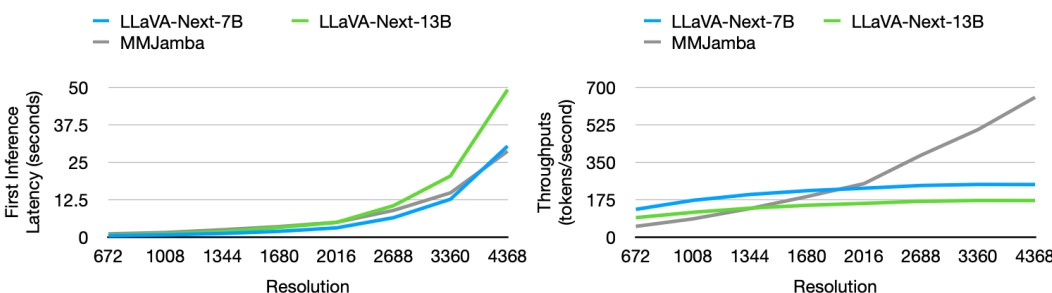

Figure 2: Efficiency analysis across different models and resolutions. First inference latency is the time used to generate 1 token and throughput is calculated as average tokens processed when generating 1024 tokens.

epoch using the AdamW optimizer and a cosine learning rate schedule, without further tuning. The learning rate is set to 1e-3 for pre-training the adapter and reduced to 7e-6 for visual instruction tuning. All experiments were performed on 32 A100 GPUs with an accumulative batch size of 256.

**Training Datasets** For two training stages of our model , we utilize high-quality data to enhance cross-modality understanding and generation. This dataset includes LLaVA-558K and Valley-702k (Luo et al., 2023b) for modality alignment during adapter pretraining and LLaVA665K, ShareGPT4V, LAION-GPT4V, DocVQA, AI2D, ChartQA, DVQA, ALLaVA-Instruct-4V and VideoChat2 (Li et al., 2024b) (about 2.7 million single- or multi-round conversations) for visual instruction fine-tuning.

**Evaluation Benchmarks** We evaluated our models on Multi-choice Video Question Answering (MC-VQA) and Open-Ended Video Question Answering (OE-VQA) to systematically assess the video understanding capabilities of our model. For the MC-VQA task, we select EgoSchema (Mangalam et al., 2023), Perception-Test (Pătrăucean et al., 2023), MVBench (Li et al., 2024b), and VideoMME (Fu et al., 2024b). We report the top-1 accuracies for all benchmarks. For open-ended question answering, we conduct comparative studies using the MSVD-QA (Xu et al., 2016) and ActivityNet-QA (Yu et al., 2019) benchmarks. Following the protocols of Maaz et al. (2024), we employ a GPT-assisted evaluation to assess the quality of the generated answers. Specifically, GPT-3.5 provides a binary "Yes-or-No" decision on the correctness of answers, and we report the percentage of "Yes" responses as Accuracy.

## 5.2 MAIN RESULTS

**Results on MC-VQA** The overall performance on multiple-choice video question answering (MC-VQA) tasks are summarized in Table 3. MMJAMBA demonstrates strong performance compared to open-source models and shows competitive results against proprietary models in certain benchmarks. For MC-VQA, MMJAMBA exhibits substantial improvements over open-source models. On the EgoSchema benchmark, MMJAMBA achieves an accuracy of 58.74%, outperforming the previous SOTA VideoLLaMA2-Mixtral (53.3%) by a large margin. Similarly, on the Perception-Test and MV-Bench datasets, MMJAMBA attains accuracies of 55.8% and 60.99%, respectively, surpassing other open-source models. Notably, MMJAMBA also outperforms the proprietary model GPT4-V on both EgoSchema and MVBench dataset. Additionally, MMJAMBA shows competitive performance on the VideoMME benchmark with an accuracy of 50.1%, highlighting its robust capabilities in video understanding tasks. It should be noted that our model outperforms the LongLLaVA (Wang et al., 2024) by a large margin on MVBench and obtain a comparable performance on VideoMME with 16 times fewer frames. As shown later, when we increased the number of frames from 8 to 32, our model's performance on VideoMME is improved to 51.8, which is better than the performance of LongLLaVA.

**Results on OE-VQA** The performance on Open-Ended Video Question Answering (OE-VQA) tasks is summarized in Table 3. MMJAMBA demonstrates strong performance compared to both proprietary and open-source models across several benchmarks. For the MSVD dataset, MMJAMBA gets an accuracy of 73.7% with a score of 4.1, outperforming other open-source models by a large margin, e.g., LLAVA-NeXT-Video (67.8%/3.5) and VideoLLaMA2-Mixtral (70.5%/3.8). However,

Table 4: Resolution distribution of different benchmarks.

| Resolution | GQA | LLaVA Wild | MM Vet | MME | vqav2 | POPE | MMMU val | SEED IMG | VisWiz | TextVQA | mmbench EN | SQA IMG |
|---|---|---|---|---|---|---|---|---|---|---|---|---|
| 336*336 | 94 | 0 | 17 | 304 | 478 | 0 | 254 | 0 | 74 | 0 | 1157 | 962 |
| 672*672 | 12484 | 12 | 79 | 966 | 106916 | 8910 | 386 | 206 | 794 | 51 | 3220 | 1004 |
| 1344*1344 | 0 | 32 | 89 | 736 | 0 | 0 | 220 | 12428 | 3091 | 4944 | 0 | 51 |
| 2688*2688 | 0 | 12 | 29 | 160 | 0 | 0 | 40 | 1266 | 4041 | 3 | 0 | 0 |
| higher | 0 | 4 | 4 | 208 | 0 | 0 | 0 | 333 | 0 | 2 | 0 | 0 |

Table 5: Performance comparison of various models on different resolutions. Res. refers to the max resolution used during inference.

| Methods | Res. | MME | MMB EN | MM Vet | LLaVA Wild | SEED IMG | MMMU val | SQA IMG | VQAT | GQA | POPE | VQAv2 | Vizwiz |
|---|---|---|---|---|---|---|---|---|---|---|---|---|---|
| LLAVA-NeXT-7b | $672^2$ | 1519 | 67.4 | 43.9 | 81.6 | 70.2 | 35.8 | 70.1 | 64.9 | 64.2 | 86.5 | 81.8 | 57.6 |
| LLAVA-NeXT-7b | $1344^2$ | 1246 | 67.4 | 30.9 | 30.0 | 20.2 | 33.0 | 68.5 | 19.4 | 64.1 | 86.5 | 81.8 | 6.07 |
| LLAVA-NeXT-7b | $2688^2$ | 1158 | 67.4 | 22.4 | 14.2 | 5.21 | 31.9 | 68.5 | 0.50 | 64.1 | 86.5 | 81.8 | 6.06 |
| LLAVA-NeXT-13b | $672^2$ | 1575 | 70 | 48.4 | 87.3 | 71.9 | 35.3 | 73.6 | 67.1 | 65.4 | 86.2 | 82.8 | 60.5 |
| LLAVA-NEXT-13b | $1344^2$ | 1353 | 70 | 35.6 | 32.8 | 32.9 | 32.4 | 72.6 | 25.4 | 65.4 | 86.2 | 82.8 | 6.51 |
| LLAVA-NEXT-13b | $2688^2$ | 1251 | 70 | 24.1 | 16.8 | 7.57 | 30.9 | 72.6 | 1.15 | 65.4 | 86.2 | 82.8 | 6.47 |
| MMJAMBA | $672^2$ | 1655 | 80.9 | 51.6 | 83.9 | 72.8 | 44.4 | 77.3 | 70.7 | 64.7 | 87.8 | 82.6 | 57.6 |
| MMJAMBA | $1344^2$ | 1647 | 80.9 | 53.1 | 82.2 | 72.8 | 44.8 | 77.0 | 71.2 | 64.7 | 87.8 | 82.6 | 56.0 |
| MMJAMBA | $2688^2$ | 1640 | 80.9 | 53.1 | 80.0 | 72 | 44.6 | 77.0 | 70.7 | 64.7 | 87.8 | 82.6 | 54.3 |

Table 6: Performance comparison of our model on different samples frames. Frame. refers to the number of frames used during inference.

| Methods | Frame. | EgoSchema | Perception | MVBench | VideoMME | MSVD | ActivityNet |
|---|---|---|---|---|---|---|---|
| MMJAMBA | 8 | 58.7 | 55.8 | 61.0 | 50.1 | 73.7/4.1 | 48.6/3.5 |
| MMJAMBA | 16 | 58.42 | 55.95 | 61.3 | 51.7 | 74.2/4.1 | 48.3/3.4 |
| MMJAMBA | 32 | 57.23 | 56.04 | 60.83 | 51.8 | 74.8/4.1 | 48.0/3.4 |
| MMJAMBA | 64 | 52.45 | 56.21 | 58.8 | 46.3 | 75.0/4.1 | 48.6/3.4 |

on the ActivityNet dataset, MMJAMBA attains an accuracy of 48.6% with a score of 3.5, which is slightly lower than LLAVA-NeXT-Video (53.5%/3.2).

## 6 ANALYSIS

**Efficiency.** In Figure 2, we present a comparative analysis of our approach against leading open-source methods under various conditions, including different resolutions and parameter counts. We focus on two aspects: first inference latency and throughputs. We measure time for generating outputs of 1 token and 1024 tokens. First inference latency is the time used to generate 1 token and throughput is calculated as average tokens processed when generating 1024 tokens. The results indicate that for both LLaVA-NeXT-7b and LLaVA-NeXT-13b, an increase in the maximum inference resolution significantly raises first inference latency but maintains a stable throughput, highlighting the inefficiency of previous leading methods. In contrast, our proposed model exhibits a markedly smaller increase in first inference latency with rising maximum inference resolution. More importantly, our model demonstrates a larger increase in throughput as the maximum inference resolution increases. Notably, at an inference resolution of $4368^2$, our model achieves a throughput that is about four times greater than that of LLaVA-NeXT-13b, which would even larger when the inference resolution is higher.

**Resolution.** In Table 5, we present the performance of our approach under various image resolutions during inference. Overall, we observe that increasing the image resolution during inference enhances the performance of our proposed model across several benchmarks. Specifically, as the resolution increases from $672^2$ to $1344^2$, there is a notable improvement in performance on benchmarks such as MM-Vet, MMMU-val, and TextVQA. For most benchmarks, including MMB-EN, SEED-IMG, GQA, POPE, VQA-v2, and SQA-IMG, performance remains consistent with higher resolutions. However, some benchmarks, such as MME, LLaVA-wild, and Vizwiz, exhibit a slight decline in performance when the resolution increases. We attribute these performance variations to

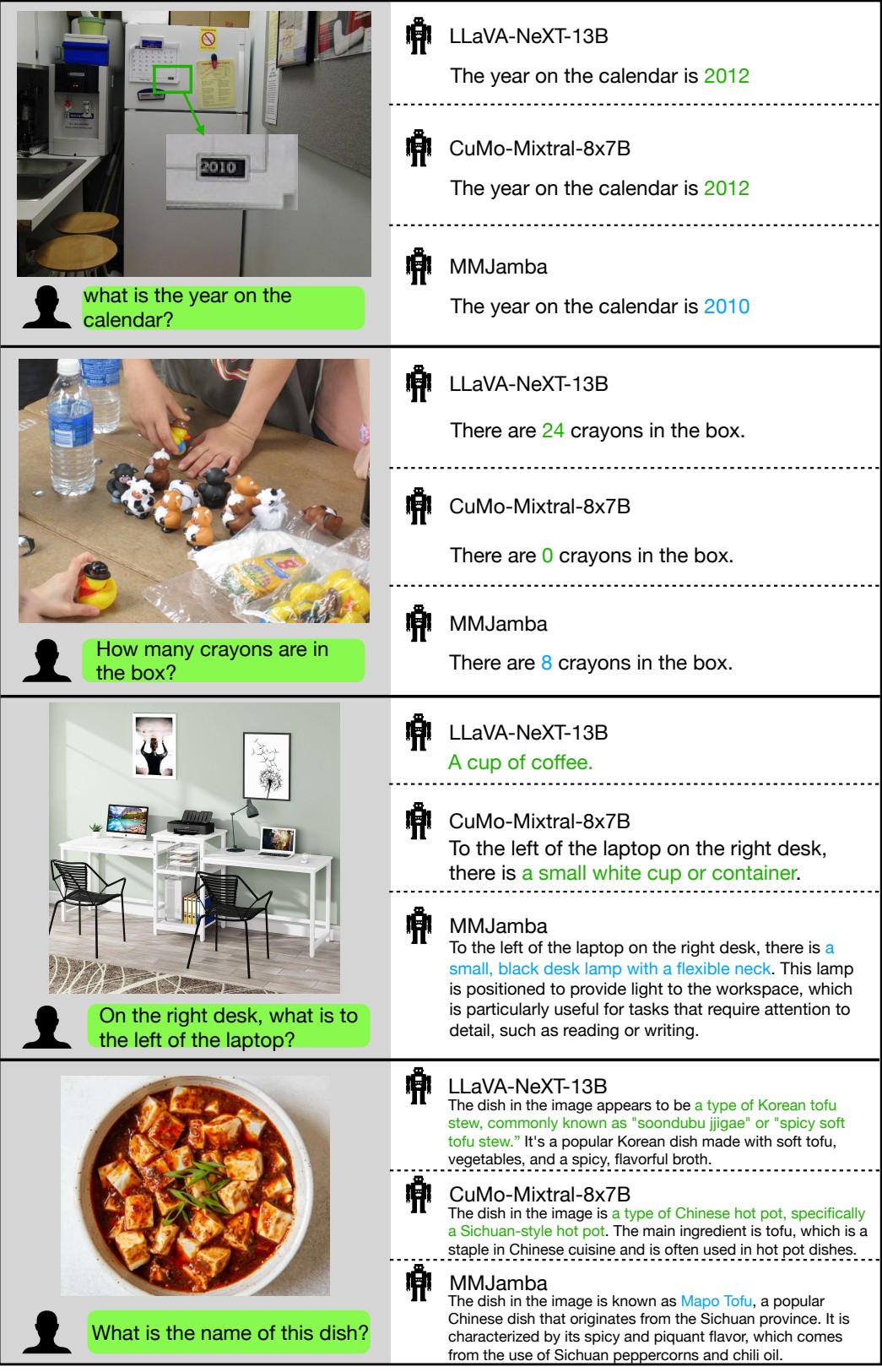

Figure 3: Dialogues between the user and MLLMs. Examples on videos are presented in Sec A.2.

the inherent nature and characteristics of different benchmarks, which may include factors such as the number of images at varying resolutions and the reliance on fine-grained visual information.

Table 7: Performance comparison of various models on different resolutions without AnyRes. Res. refers to the max resolution used during inference.

| Methods | Res. | MME | MMB EN | MM Vet | LLAVA Wild | SEED IMG | MMMU val | SQA IMG | $VQA^T$ | GQA | POPE |
|---|---|---|---|---|---|---|---|---|---|---|---|
| LLAVA-NeXT-7b | $672^2$ | 1531 | 76.0 | 44.0 | 69.4 | 70.2 | 36.4 | 70.0 | 61.3 | 64.1 | 87.4 |
| LLAVA-NeXT-7b | $1344^2$ | 6 | 3.3 | 4.9 | 3.6 | 0 | 22.7 | 7.2 | 0.48 | 0.27 | 50.6 |
| LLAVA-NeXT-7b | $2688^2$ | 0 | 0 | 3.3 | 9.4 | 0 | 25.1 | 0 | 0.06 | 0 | 50.5 |
| LLAVA-NeXT-13b | $672^2$ | 1567 | 77.8 | 49.4 | 76.7 | 70.2 | 36.0 | 73.3 | 64.3 | 65.6 | 87.3 |
| LLAVA-NEXT-13b | $1344^2$ | 7 | 4.0 | 6.7 | 4.1 | 0 | 23.9 | 8.7 | 0.55 | 0.31 | 50.6 |
| LLAVA-NEXT-13b | $2688^2$ | 0 | 0 | 0 | 10.9 | 0 | 26.0 | 0 | 0.01 | 0 | 50.5 |
| MMJAMBA | $672^2$ | 1652 | 80.8 | 52.5 | 82.7 | 72.8 | 44.3 | 75.4 | 70.6 | 64.6 | 87.8 |
| MMJAMBA | $1344^2$ | 1612 | 79.4 | 48.9 | 78.4 | 72.1 | 44.3 | 76.0 | 70.4 | 63.5 | 87.5 |
| MMJAMBA | $2688^2$ | 1417 | 73.4 | 33.2 | 72.8 | 67.8 | 40.9 | 73.3 | 65.1 | 59.4 | 83.6 |

**Frames.** In Table 6, we present the performance of our approach under various video frames during inference. Overall, we observe that increasing the sampled frames during inference enhances the performance of our proposed model across several benchmarks including Perception, VideoMME and MSVD. However, other benchmarks, such as EgoSchema, MVBench and ActivityNet, exhibit a slight decline in performance when the number of frames increases. We attribute these performance variations to the inherent nature and characteristics of different benchmarks, which may include factors such as the number of videos at varying length and the reliance on visual informations from more frames.

**Training Recipe.** In Table 5, we present the performance of our proposed approach utilizing a train-short-inference-long methodology. During training, the maximum resolution of input images is set to $672^2$, which allows for shorter sequences and consequently reduces training time. For inference, we experiment with three different maximum resolutions for input images: $672^2$, $1344^2$, and $2688^2$. To ensure a fair comparison, we evaluate our approach alongside previous leading open-source methods, all following the same train-short-inference-long strategy. Our results indicate that as the resolution of images during inference increases, the performance of baseline methods significantly deteriorates on approximately half of the benchmarks, including MM-Vet, LLaVA-wild, SEED-IMG, MMMU-val, TextVQA, and VizWiz. For other half of the benchmarks, they do not contain images with resolution higher than $672^2$. Therefore, the performance of baselines on these benchmarks will not change. In contrast, our model maintains or even improves its performance on all benchmarks with higher inference resolutions. Only three benchmarks, such as MME, LLaVA-wild, and Vizwiz, exhibit a slight decline in performance consistently when the resolution increases. This demonstrates that our proposed model can be effectively trained using lower resolutions for enhanced training efficiency while achieving superior performance when tested with higher resolutions.

# 7 CONCLUSION

In this work, we introduce MMJAMBA, a multimodal instruction tuned model utilizing the hybrid state space model–Jamba. MMJAMBA aims to effectively process long context input brought up by the higher resolutions of the images and more frames of the videos, thereby enhancing multimodal visual comprehension and recognition. We propose a train-on-short-inference-on-long recipe, which could enable our model to be trained on inputs with a short context (e.g. low-resolution images) and tested on inputs with longer contexts (e.g. high-resolution images). Extensive experiments on 12 benchmarks on images and 6 benchmarks on videos validate the efficacy of MMJAMBA. We compared our models with both open-source models, such as LLaVA-NeXT, and closed-source models, such as Gemini Pro 1.0 and GPT-4V, demonstrating the superiority of our approach. More importantly, we show that our model achieves the best efficiency when processing high-resolution images, e.g. 4 times faster than current open-source models with a comparable number of activated parameters. Additionally, we conducted extensive model analyses to understand the efficiency of our model and effectiveness of our train-short-inference-long recipe and demonstrate its capability in solving real-world multimodal tasks.

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

# A APPENDIX

## A.1 VISION ENCODING

**Image Encoder** Utilizing a static input image size for processing images or videos with varying aspect ratios, is neither efficient nor effective. To overcome this limitation, we utilize the AnyRes approach (Liu et al., 2024c), as shown in Figure 1. Our method segments the image into smaller patches, while trying to maintain the integrity of the original image's aspect ratio. More specifically, we dynamically match the optimal aspect ratio from a pre-defined set of aspect ratios. Due to limited computational resources, we allow a maximum of 4 tiles during training. Consequently, this set includes all 8 possible combinations of aspect ratios formed by 1 to 4 tiles, such as {1:1, 1:2, 2:1, 2:2, 3:1, 1:3, 1:4, 4:1}. During the matching process, for each input image, we calculate its aspect ratio and compare it with the 8 pre-defined aspect ratios by measuring the absolute difference. Then the model is sliced into multiple patches, whose resolution is same with the resolution that the vision encoder is pre-trained on. To keep the overall layout information, the raw image is also resized to the low-resolution one as the global image. Then, each image is independently encoded to a sequence of visual features by a transformer-based Visual Encoder.

**Video Encoder** For videos, we adopt a consistent frame sampling approach that extracts a fixed number of frames from each video. The extracted frames are then fed into the visual encoder. We use CLIP-ViT-Large (Radford et al., 2021), with the final layer removed, as the vision encoder. During inference, we adopt an AnyFrame mechanism which samples any number of frames from the input videos for further encoding.

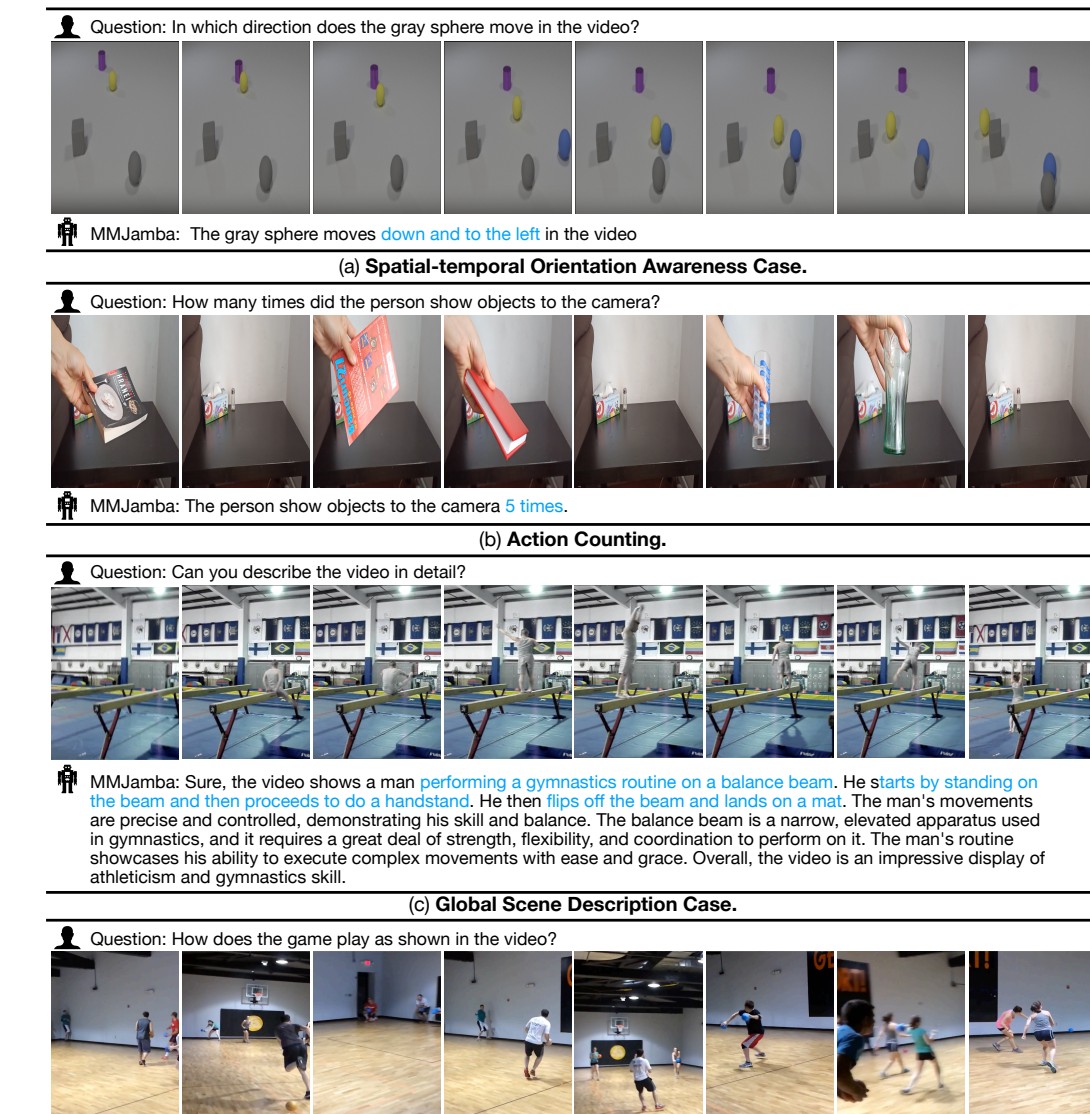

Figure 4: Qualitative Video Understanding Cases from different perspectives of MMJAMBA.

## A.2 VIDEO CASE STUDY

In Figure 4, we provide more examples of our MMJAMBA on video understanding tasks from different perspectives including spatial-temporal orientation awareness, action counting, global scene description and reasoning.

