# OpenReview forum: "Multimodal Instruction Tuning with Hybrid State Space Models"
_ICLR.cc/2025/Conference — Submitted to ICLR 2025_

### Official Review · Reviewer_DvqN · 2024-10-28

**Soundness:** 2
**Presentation:** 1
**Contribution:** 1
**Rating:** 3
**Confidence:** 4

**Summary:**

This paper aims to address the limitations of existing vision-language models in computational burdens from increasing resolution and video frame rates. They find some problems with VLMs in training long-vision token sequences. The paper does not reveal why multimodal instruction tuning specifically requires such hybrid model architectures between Transformers and state space models. They did present some good experimental results on image and video zero-shot understanding benchmarks. However, the reviewer can not find the technical novelty of this paper. Please see the comments below:

**Strengths:**

This paper shows some good results compared with baseline models. This paper contains both image and video understanding results, which provides a better evaluation compared with existing models.

**Weaknesses:**

1. Problems with resolution. Typically, we use the dynamic resolution as the key to address such limitations. Since dynamic resolution not only addresses the limitation of increasing resolution, it can also handle any aspect ratio of the image. This paper just employs the AnyRes as an engineering trick to improve the performance.

2. This paper claims their motivation originates from the long-video understanding. While their experiments in Table 3 did not support their claims. The authors did not present any advantages of their method of inference speed or performances in long video understanding. For example, VideoMME has a long-video split, the comparison should be conducted on this split. Meanwhile, Figure 2 just shows the scalability of the proposed model on handling increasing resolution, there is no such proof that supports the proposed model is more suitable for video understanding, because this paper has nothing to do with temporal aggregation or cross-frame gathering. In addition, according to Figure 2, this model has a similar inference latency with LLaVA-NeXt-13B while the resolution is in the range of [672,2688].

3. In Table 7, the authors show that their model can outperform their baseline models of LLaVA-NeXt-7B and LLaVA-NeXt-13B. In my opinion, with the resolution increasing from 1344 to 2688, their baseline models show poor performances because of memory limitation. As for the proposed model, similarly, there is an obvious performance drop as increasing the resolution from 672 to 1344 and from 1344 to 2688. The authors' claim should refer to their model can improve the performance by increasing the resolution. Therefore, it's unconvincing that the proposed model can address limitations in increasing the resolution problem.

4. The method section just follows the common practice in the community. I can not find any novelty.

**Questions:**

1. Does this paper take the LLaVA-NeXt-13B as the baseline and naively re-train the models with additional SSM blocks?

2. Performance improvement by dropping the last block of vision transformers (Page 15 Appendix A.1) seems important to the proposed model. It's a trick to improve the performance while it's not ablated in the experiments.

3. Should we use the SSM for increasing resolution or video frame rate? I think dynamic resolution in Qwen2-VL is powerful enough and their corresponding 13B model shows much better results compared with this paper.

---

> ### Comment · Reviewer_DvqN · 2024-11-25
>
> I did not receive any response from the authors.

---

### Official Review · Reviewer_kLtC · 2024-10-29

**Soundness:** 3
**Presentation:** 2
**Contribution:** 2
**Rating:** 3
**Confidence:** 3

**Summary:**

This paper proposes a multimodal instruction tuned model utilizing the hybrid state space model  to effectively process long context input brought up by the higher resolutions of the images and more frames of the videos, thereby enhancing multimodal visual comprehension and recognition. A train-on-short-inference-on-long strategy is also introduced. Extensive experiments on 12 benchmarks validate the efficacy of the proposed method.

**Strengths:**

1. The topic on processing high-resolution images or high frame rate videos is crucial, which makes the motivation necessary.
2. Extensive experiments are conducted to validate the efficacy of the proposed method.
3.  "Train-Short-Inference-Long" is interesting.

**Weaknesses:**

1. The novelty is really limited. I don’t see anything particularly unique in the architecture or training methods.
2. The presentation of results in Table is a bit hasty, with no necessary bolding, equalization, etc.
3. Ablation study is missing.

**Questions:**

1. My main concern is the novelty of the method in this paper. Can the authors elaborate on the crucial improvements in the architecture or training method?
2. The necessary ablation experiments are required.

---

> ### Comment · Reviewer_kLtC · 2024-11-27
>
> No response

---

### Official Review · Reviewer_zoqs · 2024-10-31

**Soundness:** 3
**Presentation:** 3
**Contribution:** 2
**Rating:** 5
**Confidence:** 4

**Summary:**

The paper presents a MLLM that leverages a hybrid transformer-MAMBA architecture to efficiently process long input contexts. The contributions of this study are: it introduces the MMJAMBA model optimized for lengthy contexts from high-resolution inputs, a "train-on-short-infer-on-long" strategy for improved training and inference efficiency. Experimental validation demonstrates stronger performance against both open-source and proprietary models.

**Strengths:**

The use of a SSM architecture to enhance the modeling efficiency for long contexts and tokens in MLLM is intuitively motivated.

Overall, the experimental section of the study appears to be thorough and solid, both in terms of content and effort. It also confirms that the proposed model indeed improves both efficiency and performance.

**Weaknesses:**

- The primary concern of the reviewer is the possible lack of technical innovation by the authors. Although the idea of leveraging a SSM architecture to enhance the modeling efficiency for long contexts and tokens in MLLM is interesting, the reviewer notes (in sec. 3) that the authors have simply applied the existing SSM framework to the modeling and processing of images and videos without any additional technical improvements or innovation (since this is basically explored in existing research [1-5]). While the experimental section does validate some effectiveness, the real contribution from the authors might be largely limited. And the reviewer suggests clarifying these aspects, especially distinguishing them from existing related work.

- In the Related Work section, the reviewer found that the authors did not provide a necessary survey on SSM-related technical architectures and research, which might be regrettable and inappropriate, given that the backbone of this work is exactly SSM and Mamba. Have other works already explored SSM in addressing long context modeling in MLLMs, cf. [1-5]? How do their methods and technologies differ from the one proposed in this paper? What is the unique advantage of this work? These aspects should be covered in the Related Work, which the reviewer hopes the authors will complete in the revision. Therefore, the reviewer suggests that the authors promptly supplement the discussion with SSM-related research and clarify the differences.

- Overall, the content presented in detail in this paper might be insufficient. For instance, the details of the training of the model proposed (Sec. 3.3) are not adequately addressed for the reviewer. It would be more informative to see such as the training resources consumed at each training step, the time taken, GPU hours, memory usage, or convergence rates and a comparison of training efficiency between non-SSM architectures and the model proposed in this work. Please provide further detailed explanations.

- The authors should conduct direct experimental validation for "low-resolution image modeling to high-resolution image modeling", and from "low frame rate video to high frame rate video", since these scenarios have been repeatedly emphasized in the introduction as their strong motivation. The reviewer expects to see experimental validation in these areas as direct evidence:
  1) Please compare the efficiency of the proposed model at both low and high resolutions.
  2) For fairness and clarity, the authors are advised to also display the output image resolution and video frame rate of the baseline MLLM in Tables 1, 2, and 3.

- Also, the authors have not directly compared the impact of token length. The reviewer suggests adding experiments regarding the end task performance and efficiency under different lengths of visual context (both language and visual tokens).

- One potential concern is that the scalability of SSM-based architecture in MLLM has not been fully validated yet, i.e., whether SSM can exhibit emergent phenomena similar to the Transformer architecture. Of course, this point is a discussion regarding the entire SSM architecture and not just specific to this paper. However, the reviewer would like to see a discussion on this from the authors.

Overall, the reviewer is open-minded. If the authors can actively and effectively address these concerns, the reviewer would consider raising the rating.

-------

[1] VL-Mamba: Exploring State Space Models for Multimodal Learning
[2] Efficient Classification of Long Documents via State-Space Models
[3] Speech-Mamba: Long-Context Speech Recognition with Selective State Spaces Models
[4] QMambaBSR: Burst Image Super-Resolution with Query State Space Model
[5] LongSSM: On the Length Extension of State-space Models in Language Modelling

**Questions:**

1. Do the authors intend to open-source the code and meta-data?

2. In Figure 2, why did the authors only compare with LLAVA-NEXT? Have other models been compared as well?

3. How is the resolution on the X-axis of Figure 2 calculated?

4. Have the authors compared efficiency in video modeling? It seems that the reviewer has only seen a comparison of efficiency in image modeling.

---

### Official Review · Reviewer_qA2p · 2024-11-04

**Soundness:** 2
**Presentation:** 2
**Contribution:** 2
**Rating:** 3
**Confidence:** 4

**Summary:**

The paper introduces MMJAMBA, a multimodal large language model (MLLM) designed to efficiently process long context inputs, such as high-resolution images and high-frame-rate videos. It addresses the computational challenges posed by the quadratic complexity of self-attention in transformers by integrating Mamba layers, which are part of a hybrid state space model architecture. The authors propose a "train-on-short-infer-on-long" strategy, which allows the model to be trained on low-resolution inputs for efficiency and perform inference on high-resolution inputs for enhanced performance. The model compares favorably against other SoTA methods in image and video understanding, both in terms of computation and performance.

**Strengths:**

1. The hybrid transformer-Mamba architecture effectively manages long context inputs, significantly improving inference efficiency while maintaining competitive accuracy.
1.  The "train-on-short-infer-on-long" method reduces training complexity and computational cost, making it practical for training large LMMs and on high-resolution images and long videos.
1. The model demonstrates adaptability to various resolutions and frame rates, showcasing flexibility in handling diverse multimodal tasks.

**Weaknesses:**

1. The reviewer is concerned with the novelty of this paper. As far as the reviewer can tell, the only contribution and modification made in this paper is replacing a decoder-only LLM with a state-space LLM. The claimed advantage of MMJAMBA, such as computational efficiency and the "train-on-short-infer-on-long" method are rooted in the state-space LLM, not from the novel design of MMJAMBA. Other than the different choice of LMM, everything else remains canonical to standard LMMs. For example, an MLP adapter, multi-stage training, existing training datasets and benchmarks, etc. There are limited contributions or insights to make stat-space LMM work better.

1. The comparison of MMJamba to other 13B LMMs is unfair. JAMBA LLM is a 52B MoE model with 12B active. It's performance would fall between a 52B and a 12B model.

1. The performance advantage of "train-on-short-infer-on-long" is sort of overclaimed. The performance goes up on some benchmarks indeed, but goes down or remains about the same on others. Overall, the reviewer would consider it as "maintaining similar performances across different resolutions".

**Questions:**

1. In terms of writing, Tab 7 is never referenced in the paper. Also, what's the different points the authors want to make between the "Resolution" and "Training Recipe" subsections of Sec 6 Analysis?

1. Any comparisons with existing LMMs with state-space LMMs?

---

> ### Comment · Reviewer_qA2p · 2024-11-27
> **No response received**
>
> The reviewer did not receive response from the author regarding the questions and weakness raised in the review.

---

### Official Review · Reviewer_jHBY · 2024-11-05

**Soundness:** 3
**Presentation:** 3
**Contribution:** 2
**Rating:** 3
**Confidence:** 4

**Summary:**

This paper introduces a multi-modal Large Language Model (LLM) framework that integrates a hybrid state space model called Jamba. The framework is designed to handle multi-modal tasks involving language and vision inputs, such as images and videos. The overall structure is similar to the LLaVA (Large Language and Vision Assistant) architecture, consisting of a vision encoder, an MLP adapter, and an LLM network. In this case, the LLM backbone is JamBA, as opposed to LLaMA in LLaVA.

**Strengths:**

The paper is well-written and easy to understand, making it accessible to a wide audience.

The use of State Space Models (SSMs) to reduce inference costs is a reasonable and potentially beneficial approach.

**Weaknesses:**

1. Limited Novelty: The framework resembles to LLaVA, that a vision encoder, an MLP adapter, and an LLM backbone. The training pipeline also resembles LLaVA, that first vision-language alignment training for the adapter, then instruction tuning for the LLM backbone. Train-short-infrence-long is not a new technique that is used for input length extrapolation in LLMs [1]. The hybrid model structure is from Jamba. To sum, I don’t think the novelty is enough for an ICLR paper.

2. It’s better to show the inference resolution in Tables 1, 2, 3.

Refs:

[1] Ofir Press, et al. TRAIN SHORT, TEST LONG: ATTENTION WITH LINEAR BIASES ENABLES INPUT LENGTH EXTRAPOLATION. ICLR 2022.

**Questions:**

Why use Jamba instead of pure SSM structure.

---

### Meta-Review · Area_Chair_XpdF · 2024-12-16

**Metareview:**

This paper introduces a multi-modal Large Language Model (LLM) framework that integrates a hybrid state space model called Jamba. The framework handles multi-modal tasks involving language and vision inputs, such as images and videos. The overall structure is similar to the LLaVA.


All the reviewers have found the common issues.

1, The limited novelty. The only contribution and modification made in this paper is replacing a decoder-only LLM with a state-space LLM.

2, Unfair comparison with other VLMs.

3, Missing ablation studies, such as efficiency of the proposed model at both low and high resolutions.

In addition, there are no responses during the rebuttal.

The AC has checked the entire review procedure and submission and recommends the reject of this submission.

In the next submission, the authors need to solve the above issues.

**Additional Comments On Reviewer Discussion:**

No

---

### Decision · Program_Chairs · 2025-01-22

Reject